# Deciphering the Epigenetic Alphabet Involved in Transgenerational Stress Memory in Crops

**DOI:** 10.3390/ijms22137118

**Published:** 2021-07-01

**Authors:** Velimir Mladenov, Vasileios Fotopoulos, Eirini Kaiserli, Erna Karalija, Stephane Maury, Miroslav Baranek, Naama Segal, Pilar S. Testillano, Valya Vassileva, Glória Pinto, Manuela Nagel, Hans Hoenicka, Dragana Miladinović, Philippe Gallusci, Chiara Vergata, Aliki Kapazoglou, Eleni Abraham, Eleni Tani, Maria Gerakari, Efi Sarri, Evaggelia Avramidou, Mateo Gašparović, Federico Martinelli

**Affiliations:** 1Faculty of Agriculture, University of Novi Sad, Sq. Dositeja Obradovića 8, 21000 Novi Sad, Serbia; velimir.mladenov@polj.edu.rs; 2Department of Agricultural Sciences, Biotechnology & Food Science, Cyprus University of Technology, Lemesos 3036, Cyprus; vassilis.fotopoulos@cut.ac.cy; 3Institute of Molecular, Cell and Systems Biology, College of Medical, Veterinary and Life Sciences, University of Glasgow, Glasgow G12 8QQ, UK; eirini.kaiserli@glasgow.ac.uk; 4Laboratory for Plant Physiology, Department for Biology, Faculty of Science, University of Sarajevo, 71000 Sarajevo, Bosnia and Herzegovina; erna.karalija@gmail.com; 5INRAe, EA1207 USC1328 Laboratoire de Biologie des Ligneux et des Grandes Cultures, Université d’Orléans, 45067 Orléans, France; stephane.maury@univ-orleans.fr; 6Mendeleum—Insitute of Genetics, Faculty of Horticulture, Mendel University in Brno, Valtická 334, 69144 Lednice, Czech Republic; miroslav.baranek@mendelu.cz; 7Israel Oceanographic and Limnological Research, The National Center for Mariculture (NCM), P.O.B. 1212, Eilat 88112, Israel; segaln@ocean.org.il; 8Center of Biological Research Margarita Salas, CIB-CSIC, Ramiro de Maeztu 9, 28040 Madrid, Spain; testillano@cib.csic.es; 9Department of Molecular Biology and Genetics, Institute of Plant Physiology and Genetics, Bulgarian Academy of Sciences, Acad. Georgi Bonchev Str., Bldg. 21, 1113 Sofia, Bulgaria; valyavassileva@bio21.bas.bg; 10Centre for Environmental and Marine Studies (CESAM), Biology Department, Campus de Santiago, University of Aveiro, 3810-193 Aveiro, Portugal; gpinto@ua.pt; 11Genebank Department, Leibniz Institute of Plant Genetics and Crop Plant Research (IPK) Gatersleben, 06466 Seeland, Germany; nagel@ipk-gatersleben.de; 12Genomic Research Department, Thünen Institute of Forest Genetics, 22927 Grosshansdorf, Germany; h.hoenicka@thuenen.de; 13Laboratory for Biotechnology, Institute of Field and Vegetable Crops, Maksima Gorkog 30, 21000 Novi Sad, Serbia; draganavas@yahoo.com; 14UMR Ecophysiologie et Génomique Fonctionnelle de la Vigne, Université de Bordeaux, INRAE, Bordeaux Science Agro, 210 Chemin de Leysotte—CS5000833882 Villenave d’Ornon, 33076 Bordeaux, France; philippe.gallusci@inrae.fr; 15Department of Biology, University of Florence, 50019 Sesto Fiorentino, Italy; chiara.vergata@unifi.it; 16Department of Vitis, Institute of Olive Tree, Subtropical Crops and Viticulture (IOSV), Hellenic Agricultural Organization-Dimitra (HAO-Dimitra), Sofokli Venizelou 1, Lykovrysi, 14123 Athens, Greece; akapazpglou@gmail.com; 17Laboratory of Range Science, School of Agriculture, Forestry and Natural Environment, Aristotle University of Thessaloniki, 54124 Thessaloniki, Greece; eabraham@for.auth.gr; 18Laboratory of Plant Breeding and Biometry, Department of Crop Science, Agricultural University of Athens, Iera Odos 75, 11855 Athens, Greece; etani@aua.gr (E.T.); mgerakari@aua.gr (M.G.); sarri@aua.gr (E.S.); eavramidou@aua.gr (E.A.); 19Chair of Photogrammetry and Remote Sensing, Faculty of Geodesy, University of Zagreb, 10000 Zagreb, Croatia; mgasparovic@geof.unizg.hr

**Keywords:** abiotic stress, biotic stress, epigenetic, methodology, stress memory, transgenerational memory

## Abstract

Although epigenetic modifications have been intensely investigated over the last decade due to their role in crop adaptation to rapid climate change, it is unclear which epigenetic changes are heritable and therefore transmitted to their progeny. The identification of epigenetic marks that are transmitted to the next generations is of primary importance for their use in breeding and for the development of new cultivars with a broad-spectrum of tolerance/resistance to abiotic and biotic stresses. In this review, we discuss general aspects of plant responses to environmental stresses and provide an overview of recent findings on the role of transgenerational epigenetic modifications in crops. In addition, we take the opportunity to describe the aims of EPI-CATCH, an international COST action consortium composed by researchers from 28 countries. The aim of this COST action launched in 2020 is: (1) to define standardized pipelines and methods used in the study of epigenetic mechanisms in plants, (2) update, share, and exchange findings in epigenetic responses to environmental stresses in plants, (3) develop new concepts and frontiers in plant epigenetics and epigenomics, (4) enhance dissemination, communication, and transfer of knowledge in plant epigenetics and epigenomics.

## 1. Introduction

### 1.1. Stress Memory in Plants

Chromatin marks and epigenetic regulatory mechanisms, such as DNA methylation, post-translational histone modifications and noncoding RNAs, can be dynamically changed in response to environmental stimuli, and modify gene expression levels and plant phenotypes without alterations in the underlying DNA sequence. Although plant adaptability to their environment is well-studied, stress memory has remained a challenging issue to address [1]. Stress memory in plants may be regulated not only by DNA methylation, but also by post-translational modification of histones (HPTMs) together with positioning and spacing of nucleosomes, which affect the overall packaging and the accessibility of individual regulatory elements [2]. The basic units of chromatin are the nucleosomes, consisting of histone octamers of two molecules containing of histone H2A, H2B, H3, and H4, around which 147 bp of DNA is wrapped in nearly two turns. Histone post-translational modifications (PTMs) are covalent bonds that contribute to the structure and function of chromosomes. Repressive chromatin is typically enriched in H3K9 and H3K27 trimethylation in most species [3]. Chromatin-based stress memory has been well established in the model plant species *Arabidopsis thaliana* [4]. The duration of somatic stress memory can vary within the range of days to weeks, but under certain circumstances can be extended [5,6]. The verification of epigenetic origin is the main dilemma in various stress memory events. This allows the phenomenon to be stable and heritable, but independent of the shift in the DNA sequence and thus in theory reversible. Most responses to abiotic stress that include chromatin changes are temporary and rapidly return to baseline levels when non-stress conditions have been restored. Vernalization, also known as the acceleration of flowering initiation by a prolonged duration of cold temperatures, provides a classic example of epigenetic gene silencing that is environmentally regulated. The memory of vernalization is preserved after the cold has subsided for weeks to months [7]. This includes epigenetic silencing of the flowering locus gene in *Arabidopsis* by H3K27 trimethylation and other mechanisms [8,9].

### 1.2. Epigenetic Modification Linked to Priming

Multiple attempts have been made to improve stress tolerance by inducing stress memory over time. The most potentially effective example is the activation of priming responses and epigenome targeted modifications. The priming stimulus is the trigger that initiates defense priming and induces a persistently primed state of enhanced plant defense readiness [10,11]. The process by which an environmental signal prepares a plant for potential stress exposure is referred to as priming. Defense priming is a state where, in a second attack, the plant shows a faster and more robust response compared to the initial one, thus increasing its chances of survival (Figure 1) [12].

Priming acts at the phenotypic level and does not incorporate DNA sequence changes and is thus reversible [13]. A wide-range of mechanisms inducing priming includes: a mild abiotic stress, infection by pathogens, colonization of roots by beneficial microbes, treatment with natural or synthetic chemicals, application of nanomaterials, primary metabolism alteration, and perception of certain volatile organic compounds (Figure 2) [11,14,15,16]. The priming action is followed by a period of stress memory. A modified transcriptional regulatory event during which the priming stimulus induces either sustained changes in gene expression or a modified transcriptional reaction to a secondary stimulus is one potential manifestation of memory [6]. The role of chromatin changes in stress priming was first discovered by a study that examined the effect of a secondary exposure to bacterial pathogens with respect to systemic acquired resistance responses. This priming was associated with sustained changes in histone modifications at several loci that showed priming-dependent transcriptional memory after a lag period of several days [17]. Priming is an adaptive element of induced resistance and a phenomenon with a huge potential in applications to improve crop performance and yield under suboptimal conditions [11,18,19,20]. Priming by natural compounds is being viewed as an interesting alternative for sustainable agriculture, which also contributes to exploring the molecular mechanisms associated with stress tolerance [21].

Nowadays, the need for broad-spectrum resistance of crops is more urgent than ever due to increasing trends in the severity and frequency of the occurrence of different biotic and abiotic stresses driven by climate change and globalization [22,23]. Although the priming response has been reported in several species under different stresses, it is still poorly understood [24]. Recent reports have revealed the importance and dynamic engagement of epigenetic mechanisms and indicated the fascinating possibility that epigenetic changes may be the main factor in priming establishment. Elevated levels of pattern-recognition receptors and dormant signaling enzymes, transcription factor activity (HsfB1), and alterations in chromatin state were part of the complex network underlying priming [13]. Epigenetic mechanisms such as DNA methylation and histone modifications are being verified as key factors in inducing wide-range resistance to abiotic/biotic stresses as they may be carriers of stress memory and trigger immune responses [15,24,25,26]. Increasing experimental evidence suggests that epigenetic modifications are likely involved in priming phenomena against stressors [27]. Such modifications have been shown to lead to prolonged stress memory, with seed priming in particular being implicated heavily in such transgenerational stress memory events [28]. DNA methylation and chromatin organization have been proposed to have a significant impact on priming memory [26], previously thought to occur mainly through the shortening and fractionation of H3K27me3 (histone H3 trimethylated on lysine 27) islands [29]. Resistance in *Arabidopsis* to *Pseudomonas syringae* pathovar tomato bacterial strain DC3000 increased due to previous exposure to heat, salinity, or cold stresses driven by an epigenetic-dependent mechanism [30]. This response was correlated with the hyper-induction of pattern-triggered immunity marker genes (*WRKY53*, *FRK1*, and *NHL10*) as well as with increased accumulation of H3K14ac, H3K4me2, and H3K4me3, and the requirement of the histone acetyltransferase HAC1. Histone methylation and acetylation are critical regulators of primed responses that occur at specific histone residues and are correlated with the transcription of defense-related genes.

Several reports have thus far implicated H3K4me3, H3K4me2, H3K9ac, H4K5ac, H4K8ac, and H4K12ac in defense priming [24], with histone H3K4 methylation being frequently correlated with different types of somatic stress memory [25]. Histone modifications on the promoters of transcription factors, such as methylation of histone H3K4me and acetylation of several lysine residues on histones H3 and H4 (H3ac, H4ac) on the promoters of *WRKY* genes, have been suggested to promote defense gene priming following treatment with acibenzolar S-methyl (a salicylic acid analogue) or pathogen infection [3], suggesting a histone memory for information storage in the plant stress response. In addition to histone modifications, there is increasing evidence that DNA methylation regulates priming [6]. DNA (de)methylation processes are antagonistically controlled by DNA methyltransferases and DNA demethylases [31]. Hypomethylated *Arabidopsis* mutants were reported to be resistant to the biotrophic pathogen *Hyaloperonospora arabidopsidis*, whereas hypermethylated mutants were susceptible [32]. The application of acibenzolar S-methyl has also been linked to di- or tri-methylation at lysine 4 of histone H3 (H3K4me2 and H3K4me3, respectively) and to lysine acetylation of histone H3 at lysine 9 (H3K9) or at lysine 5, 8, or 12 of histone H4 (H4K5, H4K8, and H4K12, respectively) in the promoter regions of defense-related genes [33]. In the latter study, mutant analyses revealed a tight correlation between histone modification patterns and gene priming. The application of methyl jasmonate (MeJA)-induced priming on the response efficacy to mechanical wounding of a monocot (*Oryza sativa*) showed that MeJA primes plants for increased expression of defense-related genes, such as OsBBPI and OsPOX, upon wounding in *Oryza sativa* and provided evidence that MeJA modulates histone modifications in the promoter region of OsBBPI, as well as changes at genome-wide DNA methylation level [34]. In fact, several organic molecules have been shown to effectively prime plants against abiotic stress factors such as water deficit, through the promotion of jasmonate (JA) biosynthesis and enrichment of histone H4 acetylation that serves as an epigenetic switch and is directly dependent on histone deacetylase HDA6 [33]. A recent study also demonstrated the involvement of DNA methylation in improving drought tolerance in primed rice seedlings with a cycle of mild drought and rewatering treatment [34].

DNA methylation plays a critical role in certain types of innate and acquired immunity by exerting a global influence on the responsiveness of the defence-related transcriptome via predominantly trans-regulatory mechanisms [31]. *Arabidopsis* mutants impaired in mechanisms regulating DNA methylation have been reported to show increased basal resistance to (hemi)biotrophic showing constitutive priming of PR1 gene expression and demonstrating that DNA hypo-methylation primes PR1 gene induction (normally not methylated) [35]. Interestingly, biological priming triggered by beneficial microorganisms has also been shown to lead to a modified DNA methylation status, as shown in the interaction between tomato roots and *Trichoderma harzianum* [36]. More specifically, transcriptomic analysis revealed the regulation of five histone acetyltransferases (HATs) and of components of the RNA-directed DNA methylation complex, which were accompanied by cytosine methylation changes, as revealed through analysis of global DNA methylation levels. In general, HATs and histone deacetylases (HDACs) are known to participate in control of defense priming [24], with the former being typically linked with transcriptional upregulation and the latter with transcriptional repression [37]. Recent evidence has also indicated that improved stress tolerance can be achieved following a pharmacological approach through the employment of epigenetic regulation-related compounds, such as HDAC inhibitors [38]. Representative examples of such an approach include the use of Ky-2, a class I-specific HDAC inhibitor, resulting in increased acetylation of histone H4 at AtSOS1, which encodes for a plasma membrane Na^+^/H^+^ antiporter [39]. Similarly, an exogenous application of Ky-9 and Ky-72 also led to the inhibition of specific HDACs [40,41], with both approaches leading to salt tolerance in plants.

### 1.3. Epigenetic Modifications Linked to Grafting

Grafting is an ancient technique which consists of joining two plants together [42,43]. It has been used for thousands of years to improve performance of many plant species [44]. Transcriptional and epigenetic changes seem to play a central role in phenotypic changes induced by grafting [45,46]. The movement of small RNAs in grafted plants has been demonstrated in many plant species [47,48]. Mobile small RNAs were found to exert gene silencing in graft partners either through RNA-directed DNA methylation of targeted genomic loci or through degradation of the corresponding mRNA target molecule [47,49]. Transgene-derived and endogenous small RNAs showed that the 24-nt siRNAs were able to direct DNA methylation at three sites across the graft union in the genome of the recipient cells [47]. The migration of mobile siRNA signals migrating to the roots of an *Arabidopsis* graft guided broad methylation events of the recipient root genome [50].

Grafting induces significant DNA methylation changes in many plant species [45,51]. Furthermore, locus-specific changes in DNA methylation have been reported in tomato, eggplant and pepper scions using interspecific grafts inherited to the progeny [44]. Grafting between *Brassica juncea* and *B. oleracea* promoted methylation changes, and phenotypic variation on leaves and the shoot apical meristem morphology [52]. In this case, variation was transgenerational, but also reversible after several generations [53]. Furthermore, many studies have shown that grafting induces broad transcriptome changes in plants [54]. However, the number of studies at the epigenetic level is still very limited to make generalized conclusions. Moreover, most studies have hitherto focused on changes in DNA methylation promoted by grafting, but there are no reports to date on graft-induced changes in histone modifications. A focus on further investigating a potential role for histone epigenetic marks induced by grafting would be essential for better understanding of the mechanism underlying grafting-mediated changes in gene expression.

## 2. Physiological Interplay between Epigenetic Marks, Phytohormones, and Redox State Regulating Stress Responses and Memory

While many studies have described coordinated variations between epigenetic marks, mainly DNA methylation and stress responses (gene and protein expression, production of metabolites or morphological phenotypes), the integration of epigenetic events with physiological responses of plants to stresses has been a recent focus [55,56]. The theory developed by [57] proposed that “organisms often adapt by progressing the adaptation spectrum, starting with rapidly attained physiological and epigenetic adaptations and culminating with slower long-lasting genetic ones”. This suggests (i) direct interactions between epigenetics and genetics that can occur through the DNA methylation control of TEs (transposable elements) transposition inducing mutations, but also (ii) an interplay between physiological responses and epigenetics through interactions between chromatin, redox state and hormones [58]. One example comes from the study of “heterophylla” (different shaped leaves depending on environmental conditions). Modifications of abscisic acid (ABA), ethylene, and gibberellic acid (GA) concentrations or levels of DNA methylation have been reported, suggesting that this developmental plasticity would involve an interaction between hormonal balance and epigenetics. Two arguments support this hypothesis: (i) the action of hormones in balance with complex synergies and antagonisms [59] requires a decoding to set up the expression of a “coherent” genetic program. This role would be played by the chromatin and epigenetic marks; (ii) the shift in the kinetics of events between the hormonal peak (a few minutes or hours) caused by stress and the manifestation of developmental plasticity (some days or weeks) requiring cellular memory through chromatin control as already reported for vernalization [60], “priming” [11], or transgenerational phenomena [61]. In the review by [56], three major points were addressed to support the interaction between hormonal control and chromatin: (1) an effect of hormones on chromatin and vice versa; (2) their roles on developmental plasticity or robustness, i.e., by controlling the expression of cellular identity genes in the meristems [62], by stabilizing hormone-responsive gene expression at the chromatin level, or by acting as an “integration hub” of hormonal balance [55]; (3) the biological significance of this interaction during embryogenesis or in meristems for postembryonic development [56]. This last one is supported by the central role of meristems in postembryonic morphogenesis, plasticity, and memory [39], their particularities for phytohormone signaling and chromatin remodeling [63]. The first report for this crosstalk showed that PRC2 represses PIN genes (auxin transporters) in the shoot apical meristems of *Arabidopsis* mutants, where CLV3, a cell identity gene, was knocked-out [64]. Recent studies on the shoot apical meristems of sugar beet during vernalization [65] or poplars subjected to drought or cold stress [30] showed that genes differentially expressed under the control of DNA methylation were involved in the pathways of phytohormones.

Overall, epigenetics may act as a hub between non-genetically inherited environmentally induced variation in traits and the genetically encoded traits over generations. Lastly, epigenetics can participate to memorize stress responses at the individual level (priming) or be transmitted to populations by sexual reproduction or clonal propagation (adaptation). While the interaction between phytohormones and chromatin may control plant developmental plasticity, their respective contributions and interactions are still unclear [56]. It is also possible that the redox cell state participates in this complex interaction [58]. Indeed, redox states by interacting with other signaling or hormonal pathways affect gene expression and epigenetic mechanisms [58]. Further studies are needed to improve our knowledge on such a complex interaction, in order to develop new applications for crop improvement. It will also be important to estimate the role of TEs in the response to environmental variations.

## 3. The Impact of Metabolism on the Epigenetic Regulation of Plant Stress Responses

Several reports have shown that one of the earliest response of plants to stresses occurs at the metabolic level that includes both primary and secondary metabolites, as illustrated by the profound metabolic changes that occur in response to both abiotic and biotic stresses. Indeed, epigenetic regulations are deeply connected to the metabolic status of cells, as epigenetic modifications are catalyzed by enzymes that necessitate precursors and cofactors for their activities [66]. For example, recent studies [67] indicate that there are 17 different types of enzymatic reactions that can generate histone posttranslational modifications (HPTMs) including acetylation, methylation, or phosphorylation and require various types of metabolic precursors including, acetyl Coenzyme A (acetyl CoA), S-adenosyl methionine (SAM) [68]. Additionally, histone deacetylase of the sirtuin family requires NAD+ as a cofactor, and histone phosphorylation is indeed directly connected to the energetic metabolism of cells.

Among the metabolic precursors, acetyl CoA has been the focus of attention as it is involved in the synthesis of key amino acids, lipids, and secondary metabolites and is required for histone acetylation. Acetyl CoA is produced from pyruvate, citrate, acetate, or fatty acids in various subcellular compartments of plant cells, such as chloroplasts, peroxisomes, mitochondria, cytosols, and nuclei. In particular, the acetyl CoA involved in histone acetylation is located within the nucleus. However, studies with mutants affected in organelle or cytoplasmic acetyl CoA biosynthesis pathways showed an impact of this metabolite on the epigenetic landscape of plants. For example, mutations in the gene encoding the cytosolic isoform of acetyl CoA carboxylase (ACC1) leads to an increase in acetyl CoA concentration and to histone H3 K27 hyperacetylation [69], whereas impairing the peroxisomal gene (ACX4) results in a dramatic decrease in acetyl CoA levels and histone hypoacetylation [70]. Interestingly, in the latter case, DNA methylation was also impacted at some loci normally targeted by the ROS1 DNA demethylase consistent with the close interaction between histone acetylation and DNA demethylation pathways [70]. The effect of changes in acetate availability was more specifically studied under drought stress conditions, which results in the redirection of carbon flux from pyruvate to acetaldehyde and acetate [47]. An increase in acetate content resulted in enhanced stress tolerance associated with higher levels of histone H4 acetylation and stimulation of JA synthesis. This effect was mimicked by the external application of acetate in *Arabidopsis* and other plants [47], as well as by stimulating acetate synthesis by metabolic engineering [71]. Labeling experiment using 14C acetic acid showed that the radioactivity was associated with histone H4, suggesting that acetic acid is transformed into acetyl CoA and subsequently used for histone acetylation. Hence, under drought stress, acetate synthesis is stimulated which results in histone acetylation most likely due to an increase in acetyl CoA content. This process should be controlled by the histone deacetylase, HAD6 [70]. In a more general way, acetyl CoA is central to many metabolic pathways in plants and is modulated by photosynthesis, plant cell respiration, and beta-oxidation, which are all affected under stress conditions [72]. Hence, in addition to the effect of HATs/HDACs that may lead to modification of histone acetylation levels, the direct modulation of acetyl coA levels generated by the impact of stresses on plant cell metabolism may also influence the level and genome wide distribution of histone acetylation, in a tissue specific manner.

Histone and DNA methylation represent additional epigenetic marks that depend on S-adenosylmethionine (SAM) availability as a general methyl donor. In plants, SAM synthesis depends on sulfate assimilation and folate metabolism, which are necessary to regenerate SAM from S-adenosyl homocysteine (SAH) through the SAM cycle. Fluctuation of the C1 metabolism due to mutations of genes encoding different enzymes of the folate or of the methionine cycles, or using pharmacological approaches leads to alteration of both DNA and histone methylation landscapes, demonstrating a direct link between SAM availability and these two epigenetic processes. In addition, folate metabolism is strongly affected in plants under abiotic stresses. In these conditions, a downregulation of genes / proteins involved in SAM pathways can be observed, resulting in a decrease of SAM available to histone and DNA methylation [73]. More recently, the higher susceptibility of potato plants to potato virus Y infection at high temperature was associated with a reduced expression of genes of the methionine cycle and a decrease in the content of the associated metabolites, including SAM. This evidence could be complemented by external application of methionine. Although DNA or histone methylation were not investigated in this study, the dramatic reduction in SAM availability could lead to a limitation in these epigenetic mechanisms and contribute to the increased susceptibility to virus infection, together with other regulatory processes [74]. 

Consistent with the role of the 1C metabolic pathway on plant immunity through epigenetic regulatory pathways, a mutation in the METS1 gene (Methionine Synthase), that disrupts the methionine cycle (thereby the availability of SAM) stimulates the resistance of *Arabidopsis* to *P. syringae*, whereas its upregulation results in an increased sensitivity is associated with genome-wide hypermethylation [75]. Hence, the 1C metabolism seems an essential component of the regulation of methylation-dependent epigenetic processes that play an important role in the response of plants to both abiotic and biotic stresses. Indeed, several other metabolic intermediates are likely affected in stress conditions that could interfere with epigenetic regulatory pathways, including the ATP/ADP ratio, or the NA+/NADH balance, further proving that metabolism regulation and epigenetic processes are interconnected.

## 4. The Duration of Epigenetic Stress Responses

Plants respond to stress exposure by changing the expression of a wide range of genes [76,77,78,79]. When plants are primed, they memorize the obtained experiences and are better prepared for a recurring event by rapid and/or strong induction of responsive genes. The memory does not have to be permanent, but it must last for a longer period compared to the original impulse that established the mark. Most stress memory marks are mitotically heritable [39], thus termed “somatic memory” [14]. On the other hand, stress responses that are memorized for several hours/days and disappear in return of favorable conditions are referred to as “somatic memory” [39,80]. Meiotically heritable marks form the transgenerational memory [81] that is transmittable to the next sexual generation. Maintenance of stress memories are mediated by epigenetic factors such as DNA methylation and histone modification. The complete molecular network of biological responses to environmental changes remains largely unexplored, but DNA methylation has recently been proposed to play crucial roles in rapid environmental adaptation [82]. Once the stress is relieved, the DNA methylation state can be reset to the basal level, but some DNA methylation-mediated changes can be involved in somatic or transgenerational stress memories.

The stability of DNA methylations after stress subsiding varies between species and applied stress environments. Most studies showed that DNA methylation repressed stress responsive genes under optimal growth conditions are maintained only after the stress exposure. The transcriptional activity of many stress-responsive genes is induced by DNA hypomethylation [83]. In heat-treated oilseed rape seedlings, for example, DNA demethylation events occurred in the heat-tolerant genotype, while an increase in DNA methylation occurred in the heat-sensitive genotype [84]. Approximately 40% of the observed methylation changes in grapevine plants exposed to stress by in vitro cultivation were reverted after one year, thus acting as a temporary and reversible stress acclimation mechanism. The remaining 60% of DNA methylation diversity was maintained and most likely corresponds to mitotically inherited epimutations [85]. Similar analysis focused on rice drought stress showed that 30% of the sites in which epigenetic changes occur are maintained even after the stress recovery [86]. However, during the recovery from phosphorus starvation, DNA methylation levels of stressed rice plants were still unaffected after 3 days and even after 31 days. Hierarchical clustering revealed that these samples were still more closely related to the variants with long-term P starvation than those with normal P fertilization [87]. Furthermore, when ripe tomato fruits were exposed to a chilling period, the average cytosine methylation rates increased in the promoter regions of gene loci after eight days of chilling and returned after one day of recovery to levels similar to ones before the stress. The precise genomic location and duration of DNA methylation can vary. The rate and periodicity of DNA methylation and demethylation of selected promoters was firstly described by [88] and was estimated to be approximately 100 min in human cells. Cyclic DNA methylation patterns on selected genomic regions in plants were represented for CycD3-1 gene in tobacco cells after exposure to moderate heat stress. One day after heat stress, this region shows a hypomethylation, while its methylation state returns to the basal control levels three days after the heat exposure [89].

In contrast to DNA methylation, the correlation between histone modifications and somatic stress memory is barely understood. Studies focusing on dehydration/heat stress memory [90] showed that *Arabidopsis* memory genes in conjunction with levels of Ser5P Pol II and H3K4me3 persisted for five days after the absence of drought exposure and were reduced after seven days under well-watered conditions. The accumulation of H3K4 is also involved in heat acclimation of *Arabidopsis* plants. The levels of H3K4 were elevated upon recurring heat stress, leading to hyperinduction of responsive loci and acquired thermotolerance [2]. In response to salinity stress, genome-wide analysis of chromatin modifications identified thousands of regions with differential levels of H3K4me3 or H3K27me3 in roots when compared with the control. Most of these changes sustained for 10 days after plants were returned to their control growth condition [28].

The presence of stress effects during the first stress-free generation provides an indication of intergenerational memory [90,91,92,93], whereas their presence in at least two stress-free generations is termed transgenerational stress memory [26,94,95,96,97]. The capacity of plant memory is likely operated by a molecular and biochemical machinery, and the regulation of epigenetic modifications is considered to be the most probable means for mediating inheritance [98,99]. Although transgenerational stress memory is not a general plant response and most of the environmentally induced epigenetic changes do not pass on to the offspring due to meiosis [86,100,101], more evidence is accumulating regarding stable and heritable environmentally induced epigenetic changes. Such changes can occur at any stage of the plant life-cycle and do not depend on developmental progression and dynamics [102].

Recent advances in understanding the mechanisms of transgenerational stress memory can considerably expand its potential application to crop breeding [103,104,105,106,107]. Some lines of evidence indicate that epigenetic marks have different capacities of long-term inheritance [108]. DNA methylation marks are more stably transmitted through mitosis and meiosis as compared to other epigenetic regulators, thereby they are better candidates for potential transgenerational inheritance of beneficial crop traits [79,109]. On the contrary, post-translational histone modifications are more likely to be reset during meiosis [109,110], which make them less suitable for breeding purposes. However, the epigenetic regulators are not mutually exclusive and may work together to ensure transgenerational effects [61,111]. For instance, transgenerational memory can be induced by increased temperature-induced growth through the inhibition of post-transcriptional gene silencing and attenuated plant immunity mediated by a coordinated epigenetic regulation driven by histone demethylases, heat shock transcription factors, and *trans*-acting siRNAs biogenesis [112,113].

Studies on mutants that are oppositely affected in DNA methylation display different responses to biotic stress, and do not generate long-term resistance against the pathogen [20,32]. DNA methylation/demethylation machinery modulates the defense-related transcriptome, as many of differentially expressed genes associated with the pathogen inoculation are under direct or indirect control of DNA methylation-related systems [32]. Another study has also demonstrated transgenerational effects of pathogen exposure that can be maintained over two stress-free generations [36,114]. Although the epigenetic mechanisms behind these effects remain elusive, the inheritance of pathogen resistance is often associated with DNA demethylation of transposable element sequences, and genome-wide changes in DNA methylation [114,115].

The generation of novel epigenetic patterns in *Arabidopsis* recombinant inbred lines derived from a cross between homozygous DNA methylation mutants and wild-type demonstrated that methylation variants can be trans-generationally inherited and affect several complex traits, such as flowering time, plant biomass and height, and salt stress tolerance [116,117]. This strategy has been currently applied to economically important crop species like tomato, wheat, and rice [118].

Although technological improvements have made it possible to explain aspects of plant memory thanks to a huge amount of experimental output data, some key questions still remain unanswered. What triggers the resetting of epigenetic marks or memory erasing? What drives some epigenetic marks to be mitotically and others meiotically transmissible? Particularly tempting is the notion of artificial extending of this memory in the context of the plant adaptation to the stresses that are accompanied with climate changes.

## 5. Epi-Breeding Strategies

### 5.1. Transgenerational Stress Responses and Crop Epi-Breeding

Epigenetic breeding is a powerful approach for the assessment of epigenetic marks across generations and trait improvement in crop plants [93]. The potential of epigenetic breeding has been successfully utilised in soybean through novel epigenetic variation induced by the suppression of the nuclear-encoded *MutS HOMOLOGUE 1 (MSH1*), which lead to yield improvement recorded for at least three generations. The epi-population generated by crossing the *msh1* mutant to wild type possesses a wide variation of multiple yield-related traits in greenhouse and field conditions [99]. The *MSH1* system has been also employed in *Arabidopsis* and tomato to introduce rootstock epigenetic variation in grafting experiments. In tomato, the *msh1* grafting-enhanced growth vigor in the field can be heritable over five generations illustrating the high agricultural potential of this observation [119]. Wild-type plants grafted on *msh1* rootstock exhibit enhanced growth vigor and seed yield, which are dependent on the RdDM pathway involving the function of HDA6 and MET1 [120,121]. Bisulfite sequencing and transcriptomic analysis in graft progenies have shown methylation repatterning and gene expression changes in stress and hormonal pathways. The methylome reprogramming mediated by the MSH1 RNAi transgene can contribute to phenotypic plasticity, and potentially provide increased adaptation to changing environments [121].

Transgenerational responses are often adaptive and mitigate stress damages for a better performance of crop offspring [122]. Exposure of successive rice generations to drought stress mediates improved drought adaptability, which is linked to transgenerational epi-mutations and transmission of changed DNA methylation profiles to unstressed progeny [123]. Heavy metal stress induces heritable changes in gene expression and DNA methylation in rice [124]. Progeny of heat stressed *Brassica rapa* displays changes in the abundance of transfer RNA fragments (tRFs) and small nucleolar RNA fragments (snoRNA), which are associated with improved plant performance to future adverse conditions mediated by epigenetic and physiological adjustments [125]. Symmetric CG methylation was stably maintained for at least two generations upon potato spindle tuber viroid infection in tobacco, independently of RdDM [126].

### 5.2. Molecular Epi Breeding

Natural, transgenic, environmental, or chemically induced epigenetic modifications can lead to epi-allele variants that have an effect on stress resilience. If permanently transmitted to subsequent generations, these epigenetic variants could serve as breeding targets for crop improvement. Transgenic knockdown lines of a rice microRNA, *miR166*, displayed increased tolerance to drought stress and these plants exhibited morphological changes observed in natural plant dehydration responses such as leaf rolling and decreased xylem diameter [127]. In another study, increased salinity resulted in differential DNA methylation in genomic regions of a salt-tolerant rice variety. Interestingly, these regions encompassed specific stress-responsive genes and differential DNA methylation observed within or in the vicinity of these genes was associated with transcriptional activity [128]. Likewise, drought-induced differential DNA methylation was found associated with drought stress- responsive genes in rice. Notably, the differential profile of DNA methylation was maintained in drought-exposed progeny pointing to the existence of drought inducible epi-marks that may be heritable in successive generations [123]. In a recent report, substantial genome-wide reprogramming of the H3K27me3 mark was found to promote tillering and higher yield in rice even under reduced nitrogen inputs [129,130]. Epigenetic manipulation of TEs may lead to desirable expression changes in nearby genes associated with environmental responses and adaptation [131,132]. Heat-inducible TE mobilization in rice may be achieved in the same manner that *ONSEN* retrotransposons have been mobilized in heat-stressed *Arabidopsis* and stably transmitted to progeny generating variability that could be used in breeding programs [133]. Epigenetic variation has been also found in relation to biotic stresses. DNA methylation changes were evidenced in rice infected with the rice blast fungus *Magnaporthe oryzae*. Differentially methylated regions across infection stages were enriched in genes involved in responses to external stressors and cell communication, including NBS-LRR families, and MYB and WRKY transcription factors [134].

In maize, a TE insertion into the promoter of the ZmCCT gene, which is involved in the regulation of the photoperiod response in maize, repressed gene expression and led to decreased photoperiod sensitivity allowing for domestication spread of maize to long day environments [134]. In addition, GWAS in maize revealed that a MITE (miniature inverted repeat transposable element) insertion in the promoter of a NAC transcription factor encoding gene was associated with variation in drought tolerant maize phenotypes and may be exploited for genotype selection for drought tolerance [135]. A TE-derived siRNA from a MITE insertion in the intron of a *WRKY45* transcription factor gene was found to target the *ST1* gene, a major determinant for resistance to the devastating pathogene *Xanthomonas oryzae pv. oryzae* (Xoo), resulting in RdDM related repression of the gene and attenuation of resistance to Xoo [136].

The authors of [137] determined differentially methylated profiles between high and low synchronous pod maturity (SPM) groups in a mungbean recombinant inbred line (RIL) population. The differentially methylated regions (DMRs) were independent of genetic variation and significantly affected the expression of nearby genes. The genes correlated with these SPM-associated genic DMRs were enriched for transcription factors such as bZIP, AP2, and ARF, related to auxin, ABA, and ethylene-signal transduction pathways, as well as four proteins involved in the gibberellin transduction pathway. These findings indicate that the SPM phenotype is established through epigenetically regulated hormonal pathways and point to the utilization of the respective epiallele variants as a breeding resource for crop improvement in mungbean and other related crops.

The manipulation of the *MSH1* plant-specific gene was utilized for inducing epigenetic variation associated with agronomical traits in soybean [119]. *MSH1* epi-populations were generated by crossing msh1-acquired memory lines (produced by RNAi suppression of *MSH1*) to wild-type plants. Derived soybean epi-lines exhibited a wide range of phenotypic variation in yield-related traits such as pods per plant, seed weight, days to flowering, and maturity time in both glasshouse and field trials. Selected lines displayed enhanced seed yield and differential expression for genes associated with enhanced growth across generations, including genes related to cell cycle, ABA biosynthesis, and auxin response. These results suggested that MSH1-driven epigenetic variation may be exploited for developing improved varieties with enhanced yield and stability.

### 5.3. Epichemicals in Breeding

Knowledge gained in recent years has revealed that epigenetic regulation plays a key role during in vitro regeneration and propagation [138,139]. Global changes of key epigenetic marks, particularly the reduction of DNA methylation, H3K9me2, H3K27me, and the increase of histone acetylation, are required to trigger plant cell reprogramming to embryogenesis, while increasing levels of these epigenetic modifications regulate embryo differentiation, in several model and crop species [60,140,141]. Epigenetics chemical targeting has shown high potential in reprogramming cancer cells and human infections. Also in plant biology research, chemical approaches with epigenetic modulators have been successfully used to decipher epigenetic molecular pathways and to improve in vitro plant regeneration/propagation yield, through organogenesis and embryogenesis. Epigenetic inhibitors that promote global DNA and H3K9 demethylation and histone acetylation have shown to enhance cell reprogramming and embryogenesis initiation [141,142,143,144,145,146,147,148,149,150].

Recent studies also showed that epigenetic mechanisms may have a role in increasing crop resilience to specific stresses and therefore may be an important tool for generating new, more environment-flexible varieties [107]. For example, target gene repression by small non-coding RNAs was found to be involved in drought stress response in barley, where in drought-stressed plants, the promoter region of cytokinin-oxidase 2.1 (HvCKX 2.1) had increased level of DNA methylation [145]. A high proportion of the drought-induced epimutations (DNA methylation changes) maintained their altered methylation pattern in successive generations of rice exposed to drought, indicating the presence of possible epi-marks that are drought inducible and heritable across generations [123]. Furthermore, chemical priming with epigenetic inhibitors is believed to represent a promising strategy for mitigating abiotic stress in crop plants, although further research is needed to understand epigenetic priming effects over abiotic stress resistance [39]. Exploiting epigenetic variations for breeding applications is not a straightforward process and faces different challenges when methods and approaches used in model plants are used in the crops. One of the main challenges is whether mutations in DNA methylation mechanisms are tolerated by the crop and if the produced mutants would be viable [105,146]. Consequently, the complex nature of most of the crop genomes would need to be taken into account, and more precise approaches that have been developed in recent years used, such as epi-mutagenesis and targeted epigenome editing for direct epigenome engineering, as has been described in *Arabidopsis* [103,147]. Applicability of epigenetic variations in a certain crop also depends on the way epigenetic variation is transmitted, as in some cases, it affects their relevance for differently propagated crops [108]. Finally, use of natural or induced epigenetic variations in crop breeding requires that these variations are stable and heritable, since most of the stress-induced epigenetic modifications return to initial levels when the stress is removed [82,91,148]. Hence, further study of the factors affecting epiallele stability in crops is needed along with the development of mathematical models for the increase and identification of heritable epigenetic phenotypes in order to avoid inducing epialleles that are unlikely to be stable during the breeding process [149,150].

Combined with classical genetic studies, newly available genome sequences of the main crops, along with sequencing technologies, are facilitating the study of the epigenetic phenomena at a whole genome level and pave the way for application of epi-molecular breeding in crops. This, together with the new epigenome editing tools, could generate new avenues for using the full potential of epigenetics in crop improvement [103]. Genome-wide mapping of epigenetic marks and collecting and normalizing plant epigenomic data for a range of species will enhance understanding of the role of epigenomic functions in crop response to the stress and efficient application of epi-molecular tools in crop breeding [151]. Furthermore, the design and synthesis of novel chemical libraries for epigenetic targets is rapidly growing. The increasing fruitful interactions between plant epigenetic researchers and chemical biology experts, together with development of assays for high-throughput screening of epigenetic compounds and phenotyping will lead to the creation of new chemical tools as well as translational applications that impact upon accelerated plant breeding and crop adaptation to climate change. Finally, there is a need for more data on epi-alleles and epigenetic targets in a greater range of plant species in order to gain a more comprehensive understanding of the mechanisms inducing and stabilizing epigenetic variation in crops. This will also require a multidisciplinary effort of researchers involved in different areas of plant science and crop breeding and better integration of epigenomic data obtained in different crops.

## 6. The EPI-CATCH Consortium: A Way to Gain Insight into Crop Stress Memory

As summarized in this review, evidence from model plants indicates that epigenetic modifications play an important role in the regulation of plant responses to the environment. However, most of the epigenetic studies have been carried out in model plants, particularly *Arabidopsis* and the role of epigenetic variation in adaptation of crops to abiotic stress is still not well understood (Niederhuth and Schmitz, 2014). EPI-CATCH (EPIgenetic mechanisms of Crop Adaptation To Climate cHange) (https://www.epicatch.eu, accessed on 29 June 2021) is a COST Action (CA19125) that was launched in September 2019 with the aim to define, develop, generate, and share new breakthrough knowledge and methodologies for the investigation of epigenetic mechanisms modulating crop adaptation to environmental stresses driven by climate change, thus contributing to the understanding of the role and possible applications of epigenetic variation in crop improvement. The main purpose of EPI-CATCH Consortium is to define and publicly deliver standardized methods, workflows, and research pipelines for crop epigenomics needed to allow the agriculture of tomorrow to be prepared for future stresses caused by climate change, maintaining its sustainability towards the environment and human beings. EPI-CATCH will promote close networking activities between international research groups and stakeholders in all the fields of the agricultural and forestry sector where epigenetics has an impact (Figure 3).

For further integration of epigenetics and epigenomics in crop improvement, more work needs to be done for creating new, reliable, and efficient ways to move beyond mere correlation between epigenetic variation and the desired trait [107]. Within its activities, EPI-CATCH will contribute to this integration through linking epigenetic modifications with phenotypic and genomic data to connect structural and allelic variations with key epigenetic marks in order to obtain epi-molecular markers associated with highly desired agronomic and qualitative traits in the context of climate change. Development of epi-molecular markers, and especially epi-QTLs, will further enable the use of targeted, gene-specific modifications to the epigenome that would lead to the anticipated stress responses and creation of desired phenotypes.

Use of epigenetic variations in crop improvement requires that these variations are stable and heritable [148]. However, as discussed in this review, more data on the factors affecting epiallele stability and heritability in crops are needed in order to avoid inducing epialleles that are unlikely to be stable during the breeding process [150]. In order to overcome this obstacle, two approaches could be used: development of mathematical models and/or epigeno-typing procedures [107,151]. The EPI-CATCH consortium aims to develop new instruments and mathematical models for better linking epigenomic data with those obtained from climatology, agronomy, plant stress biology, plant pathology, and molecular breeding.

Furthermore, EPI-CATCH will work on identifying an “epigenetic alphabet” in parallel with the well-established genetic code underlying crop adaptation to climate change and introducing new concepts and terms such as “epigenome plasticity” and “epigenome expansion” enabling accurate identification of epialleles of interest, as well as better understanding of patterns of their inheritance.

Finally, EPI-CATCH will contribute to a more comprehensive understanding of the mechanisms inducing and stabilizing epigenetic variation and stress memory in crops by working on a range of plant species and addressing crop epigenetics in a holistic way using inter-disciplinary techniques, strategies, and methods. By stimulating the exchange of new fundamental knowledge on crop genotype-environment interactions, epigenetic variations and stress memory among working groups of scientists from 28 different countries’ EPI-CATCH will facilitate cross-species comparisons, annotation of genomes, and understanding of the role of epigenetic variations in crop response to the stress. 

## Figures and Tables

**Figure 1 ijms-22-07118-f001:**
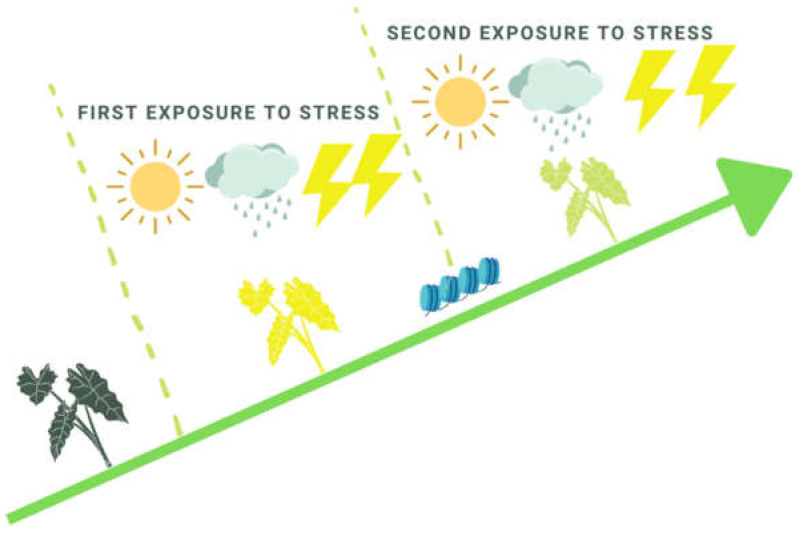
A model of plant memory acquirement due to consequential exposure to environmental stresses.

**Figure 2 ijms-22-07118-f002:**
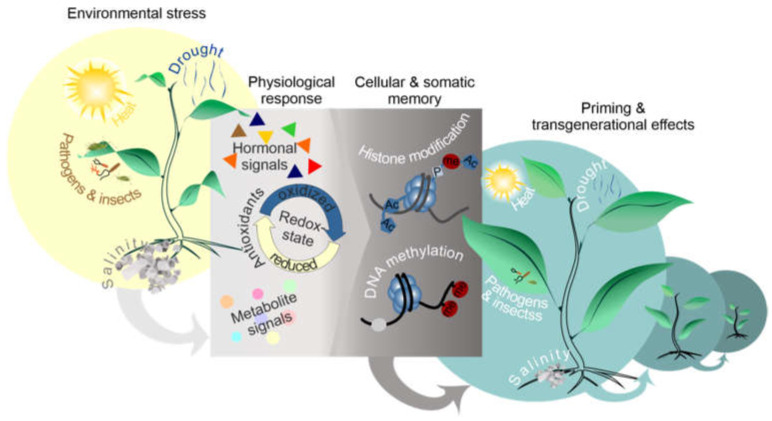
Diagrammatic overview of the environmental and physiological factors involved in priming and transgenerational effects. Priming is caused by environmental and physiological factors (antioxidants, redox state, hormonal crosstalk, metabolites signals).

**Figure 3 ijms-22-07118-f003:**
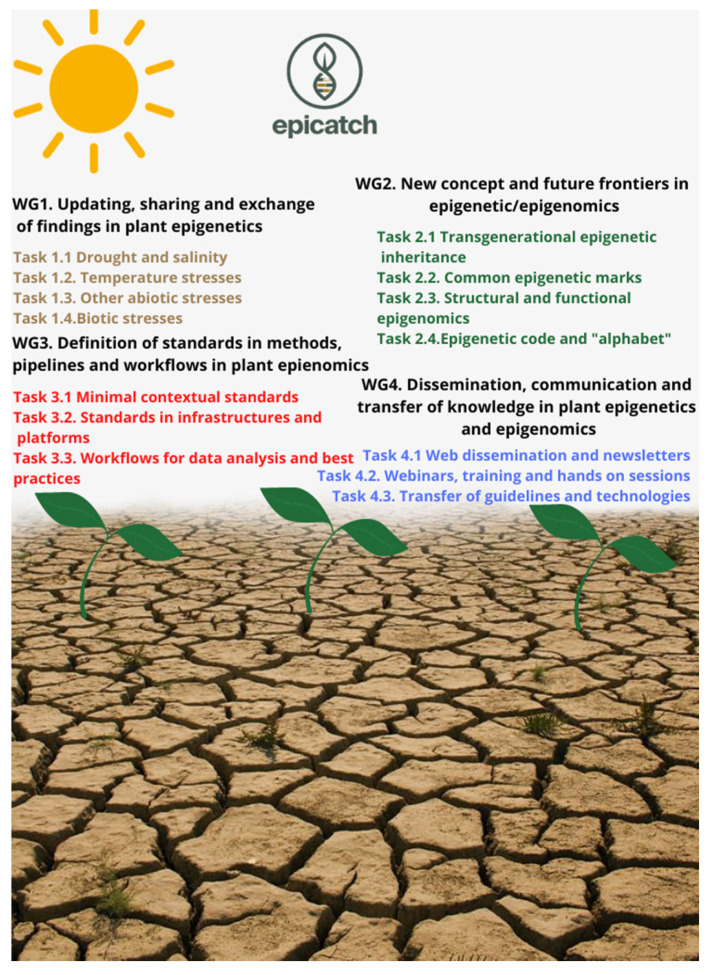
The structure of EPI-CATCH COST Action: working groups and tasks.

## Data Availability

Not applicable.

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
