# Peer review of "Deciphering the Epigenetic Alphabet Involved in Transgenerational Stress Memory in Crops"

_ijms, 2021, doi:10.3390/ijms22137118_

Round 1

Reviewer 1 Report

In general, the concept of the manuscript "Deciphering the epigenetic alphabet involved in transgenerational stress memory in crops" is a great one with many useful applications for the field of plant epigenetics.  

However, the manuscript in its current state would require a substantial amount of rephrasing/rewording in order to clearly comprehend the author's intent - certain chapters more so than others. The manuscript could also benefit from a moderate amount of English language editing.

However, there are certain aspects of the manuscript that are rather confusing:

  1.  It is not entirely clear why the authors included a section on large-scale crop stress monitoring by remote sensing techniques. It seems rather out of place in this paper. Perhaps the authors should consider removing it to keep the review concise and relevant.
  2. The brief description of the EPI-CATCH consortium and the working group plans seemed like it did not belong in this review. If the intention was to discuss EPI-CATCH, then perhaps more details should be given about the EPI-CATCH. For example, what is the current website? What are some of the possible standardized pipelines being considered? Where have these pipelines been published before? Why are they better than others? Authors could even expand the challenges of cross comparing data from different publications.

Other comments: 

Figure 3 - WG1 shows a list of very specific stresses. Why not flooding and other abiotic stresses? The plant community would greatly expand this area of research if the stresses are not so specific. If this is not the case, perhaps a clearer statement might about the stresses might be included.

Author Response

Reviewer 1

In general, the concept of the manuscript "Deciphering the epigenetic alphabet involved in transgenerational stress memory in crops" is a great one with many useful applications for the field of plant epigenetics.  

However, the manuscript in its current state would require a substantial amount of rephrasing/rewording in order to clearly comprehend the author's intent - certain chapters more so than others. The manuscript could also benefit from a moderate amount of English language editing.

Reply: We have corrected the English language with a significant rephrasing/rewording as suggested.

However, there are certain aspects of the manuscript that are rather confusing:

  1. It is not entirely clear why the authors included a section on large-scale crop stress monitoring by remote sensing techniques. It seems rather out of place in this paper. Perhaps the authors should consider removing it to keep the review concise and relevant.

Reply: we removed this part.

  1. The brief description of the EPI-CATCH consortium and the working group plans seemed like it did not belong in this review. If the intention was to discuss EPI-CATCH, then perhaps more details should be given about the EPI-CATCH. For example, what is the current website? What are some of the possible standardized pipelines being considered? Where have these pipelines been published before? Why are they better than others? Authors could even expand the challenges of cross comparing data from different publications.

Reply: Considering that EPI-CATCH Cost Action has started in September 2020, working groups are still defining which epigenetic pipelines, methods and workflow to consider in order to standardize research activity on plant epigenetics. We have provided information about our website. We have shortened this part in order to provide just a rapid outlook on ongoing activities of our consortium.

Other comments: 

Figure 3 - WG1 shows a list of very specific stresses. Why not flooding and other abiotic stresses? The plant community would greatly expand this area of research if the stresses are not so specific. If this is not the case, perhaps a clearer statement might about the stresses might be included.

Reply: We have change name of Task 1.3 specifying that we will deal with plant epigenetic responses to any abiotic stresses.

Reviewer 2 Report

Mladenov et al. is a very long review that is clear has been written by many authors simultaneously as different styles are detected along the manuscript. Except for few references that I missed during my reading and few typos, overall is a good review and support its publications. I would also reconsider the Figure design as look more for a textbook than for a scientific journal.

Author Response

Reviewer 2

Mladenov et al. is a very long review that is clear has been written by many authors simultaneously as different styles are detected along the manuscript. Except for few references that I missed during my reading and few typos, overall is a good review and support its publications. I would also reconsider the Figure design as look more for a textbook than for a scientific journal.

Reply: We have integrated all reviewer’s suggestions in the re-submitted manuscript. If possible, we would like to keep these figures because we believe they are helping to express concepts described in the review.